# Mutational Spectrum of the *ABCA12* Gene and Genotype–Phenotype Correlation in a Cohort of 64 Patients with Autosomal Recessive Congenital Ichthyosis

**DOI:** 10.3390/genes14030717

**Published:** 2023-03-15

**Authors:** Alrun Hotz, Julia Kopp, Emmanuelle Bourrat, Vinzenz Oji, Kira Süßmuth, Katalin Komlosi, Bakar Bouadjar, Iliana Tantcheva-Poór, Maritta Hellström Pigg, Regina C. Betz, Kathrin Giehl, Fiona Schedel, Lisa Weibel, Solveig Schulz, Dora V. Stölzl, Gianluca Tadini, Emine Demiral, Karin Berggard, Andreas D. Zimmer, Svenja Alter, Judith Fischer

**Affiliations:** 1Institute of Human Genetics, Medical Center—University of Freiburg, Faculty of Medicine, University of Freiburg, 79106 Freiburg, Germany; 2Center for Cornification Disorders, Freiburg Center for Rare Diseases, Medical Center, University of Freiburg, 79106 Freiburg, Germany; 3European Reference Networks (ERN Skin), 75015 Paris, France; 4Department of Dermatology, Reference Center for Rare Skin Diseases MAGEC, Saint Louis Hospital AP-HP, 75015 Paris, France; 5Department of Dermatology and Venereology, Muenster University Medical Center, 48149 Muenster, Germany; 6Department of Dermatology, CHU of Bab-El-Oued Algiers, Algiers 16008, Algeria; 7Department of Dermatology and Venereology, Faculty of Medicine and University Hospital, University of Cologne, 50937 Cologne, Germany; 8Clinical Genetics, Karolinska University Hospital, 171 64 Solna, Sweden; 9Institute of Human Genetics, University of Bonn, Medical Faculty & University Hospital Bonn, 53127 Bonn, Germany; 10Department of Dermatology, Venerology und Allergology, University Hospital of Munich, 80337 Munich, Germany; 11Pediatric Skin Center, Dermatology Department, University Children’s Hospital Zurich, 8032 Zurich, Switzerland; 12Synlab Medical Practice for Human Genetics Jena, 07747 Jena, Germany; 13Center for Inflammatory Skin Diseases, Department of Dermatology and Allergy, University Hospital Schleswig-Holstein, Campus Kiel, 24105 Kiel, Germany; 14Pediatric Dermatology Unit, Department of Pathophysiology and Transplantation, Fondazione IRCCS Ca’ Granda—Ospedale Maggiore Policlinico, University of Milan, 20122 Milan, Italy; 15Department of Medical Genetics, Inonu University School of Medicine, 44280 Malatya, Turkey; 16Department of Dermatology and Venereology, Skåne University Hospital, 221 85 Lund, Sweden

**Keywords:** *ABCA12*, ARCI, harlequin ichthyosis, lamellar ichthyosis, congenital ichthyosiform erythroderma

## Abstract

Autosomal recessive congenital ichthyosis (ARCI) is a non-syndromic congenital disorder of cornification characterized by abnormal scaling of the skin. The three major phenotypes are lamellar ichthyosis, congenital ichthyosiform erythroderma, and harlequin ichthyosis. ARCI is caused by biallelic mutations in *ABCA12*, *ALOX12B*, *ALOXE3*, *CERS3*, *CYP4F22*, *NIPAL4*, *PNPLA1*, *SDR9C7*, *SULT2B1*, and *TGM1*. The most severe form of ARCI, harlequin ichthyosis, is caused by mutations in *ABCA12*. Mutations in this gene can also lead to congenital ichthyosiform erythroderma or lamellar ichthyosis. We present a large cohort of 64 patients affected with ARCI carrying biallelic mutations in *ABCA12*. Our study comprises 34 novel mutations in *ABCA12*, expanding the mutational spectrum of *ABCA12*-associated ARCI up to 217 mutations. Within these we found the possible mutational hotspots c.4541G>A, p.(Arg1514His) and c.4139A>G, p.(Asn1380Ser). A correlation of the phenotype with the effect of the genetic mutation on protein function is demonstrated. Loss-of-function mutations on both alleles generally result in harlequin ichthyosis, whereas biallelic missense mutations mainly lead to CIE or LI.

## 1. Introduction

Autosomal recessive congenital ichthyosis (ARCI) is a heterogeneous group of disorders of cornification characterized by abnormal skin scaling over the whole body. ARCI belongs to the non-syndromic types of ichthyoses. Three major clinical phenotypes have been described for ARCI: lamellar ichthyosis (LI), congenital ichthyosiform erythroderma (CIE), and harlequin ichthyosis (HI). LI is characterized by large and dark scales without erythroderma, whereas CIE shows fine white scales with erythroderma. Collodion membrane at birth can be found in both LI and CIE. HI is the most severe form of ARCI. Affected newborns were mainly born preterm and classically show large, thick, plate-like scales covering the body with deep erythematous fissures, accompanied by characteristic facial features such as ectropium, eclabium, flattened ears, and nasal hypoplasia. Limbs with flexion contractures and hypoplastic digits often occur. HI is associated with increased perinatal mortality due to feeding problems, respiratory distress, increased water loss, electrolyte imbalances, impaired thermal regulation, and bacterial infections [1]. Intensive neonatal care can improve the survival of affected newborns. After infancy, patients show generalized red, scaly skin and often additional features such as alopecia and contractures of fingers. They are prone to hearing difficulties due to a blockade of the external auditory canal by skin residues. Different ophthalmologic problems are frequent. Some patients show developmental delay in different areas. Adults may develop inflammatory arthritis. Treatments for ARCI include emollients and other topical treatments as well as systemic agents such as retinoids, vitamin D and its analogs, anti-inflammatory drugs, enzyme replacement, and substitution therapies [2]. Erythema and disease severity highly correlate with IL-17 expression in ichthyosis. Therefore, IL-17/IL-23-targeting strategies can be another therapeutic option [3]. The IL-17A inhibitor secukinumab induced a significant reduction in erythema and scaliness and a resolution of ectropion in a patient with *ABCA12*-associated ARCI [4]. 3D models of ichthyotic skin equivalents provide new insights in dysregulated pathways as feasible therapeutic targets [5]. Gene therapy as a curative method is still at an early stage of development [6].

ARCI is caused by biallelic mutations in several genes: *ABCA12*, *ALOX12B*, *ALOXE3*, *CERS3*, *CYP4F22*, *NIPAL4*, *PNPLA1*, *SDR9C7*, *SULT2B1*, and *TGM1*. In patients with ARCI, mutations in *TGM1* are found most frequently [7]. *ABCA12* (*ATP-binding cassette*, *subfamily A*, *member 12*) was first described by Lefévre et al. [8] as a causative gene for LI. Later, pathogenic variants in *ABCA12* were described to be causative for HI [9,10,11]. Mutations in *ABCA12* can lead to HI, CIE, or LI depending on the type of the mutation. The main cause for HI are truncating mutations in *ABCA12* (ichthyosis, congenital, autosomal recessive 4B (harlequin), and MIM #242500), whereas missense mutations in *ABCA12* usually lead to milder phenotypes such as CIE or LI (ichthyosis, congenital, autosomal recessive 4A, and MIM #601277) [8,12]. The ABCA12 protein comprises 2595 amino acids and includes two transmembrane domains, each consisting of six hydrophobic membrane-spanning helices, and two ATP-binding cassettes, including highly conserved motifs (Walker A, Walker B, and active transport signature) [10]. Like many other genes implicated in ARCI, *ABCA12* is a part of the ceramide pathway in the epidermis. Ceramides are important as free lipids in the extracellular space and for building the cornified lipid envelope (CLE). The CLE is a monolayer of lipids, which are an interface between the hydrophilic corneocytes and the lipophilic extracellular lipids and are therefore important for the skin barrier [13]. The protein-bound ceramides are formed in a cascade of reactions from sphingoid bases and fatty acids. ARCI-related genes and other skin-affecting genes are involved in many intermediate steps of these reactions [14]. ABCA12 is a keratinocyte transmembrane lipid transporter protein involved in the transport of lipids in lamellar granules (LG) to the apical surface of granular layer keratinocytes [10]. In healthy skin, the LGs fuse with the cell membrane in the interface of stratum granulosum and stratum corneum and discharge the content into the intercellular lamellae thus constituting a water-permeability barrier [15]. Mutations in *ABCA12* lead to abnormal LGs or to a reduced number of LGs and an impaired transport of these granules, which results in a strong reduction in lipids in the intercellular spaces of the stratum corneum and a defective lipid barrier [12,15]. LGs originate from the Golgi apparatus, especially the trans-Golgi. Examination of isolated LGs revealed that they predominantly contain phosphoglycerides, sphingomyelin, and glucosylceramides. Furthermore, the lamellar bodies also contain enzymes such as acid hydrolases, including glucocerebrosidase, sphingomyelinase, and phospholipase A, as well as proteases and antimicrobial peptides [16].

In this study, we present a large cohort of 64 patients affected with ARCI carrying 62 different mutations in *ABCA12*. To date, 183 variants have been described as pathogenic for ARCI (HGMD^®^ Professional version 2022.4). Here, we add 34 novel mutations in *ABCA12*, which expand the mutational spectrum of *ABCA12*-associated ARCI. Furthermore, our large cohort allows new insights into genotype–phenotype correlations in *ABCA12*.

## 2. Methods

In 64 patients with ARCI, mutations in *ABCA12* were detected using different sequencing methods including Sanger sequencing or next generation sequencing (NGS). Due to the large time span of 28 years over which the patients were diagnosed, different sequencing methods were used. We present the results of the mutational analysis for only one affected individual from each family. This study was conducted according to the principles of the Declaration of Helsinki.

In all patients, genomic DNA was isolated from peripheral blood lymphocytes. Subsequently, PCR amplification, Sanger sequencing, or NGS methods through targeted multi-gene panels were performed. Most of our patients were analyzed through the targeted multi-gene panels HaloPlex Custom Kit or SureSelect Custom Kit (Agilent Technologies, Inc., Santa Clara, CA, USA). In all patients of our cohort, all coding exons and flanking intronic sequences (+/−20 base pairs) of *ABCA12* (transcript ENST00000272895.7, RefSeq NM_173076.2, protein accession number NP_775099.2, GRCh37.p13) were analyzed. The resulting data were analyzed using an in-house bioinformatics pipeline and the commercial software SeqNext (JSI medical systems).

Alignments were retrieved from Ensembl 109 [17] using Eutheria Gen Tree node. Analysis and visualization were performed with Jalview version 2.11.1.3-j1.8 (https://www.jalview.org/, accessed on December 2022) [18]. The Genome Aggregation Database version v2.1.1 [19] and the ClinVar version, 10 December 2022 [20], were used. The classification of the detected sequence variants is based on the ACMG standards and guidelines [21].

## 3. Results

Genetic analysis of patients with ichthyosis revealed 64 families with mutations in *ABCA12* in our laboratories. All patients and their mutations are listed in Table 1. We found 34 novel mutations in *ABCA12*, which are shown in bold in Figure 1A and Table 1. The classification of the novel mutations is presented in the Appendix A. All novel mutations in our cohort are classified as likely pathogenic or as pathogenic based on the ACMG standards and guidelines [21]. Our study increases the total number of known *ABCA12* mutations to 217 (HGMD^®^ Professional 2022.4).

### 3.1. Spectrum of Mutations in ABCA12

In our cohort of 64 patients, 62 different mutations were found occurring in a homozygous or compound heterozygous state, 34 of which are novel mutations. Due to the very rare occurrence of the mutations in *ABCA12*, patients with homozygous mutations usually have a consanguineous background. A total of 22 mutations of our cohort (35.5%) were missense mutations, 17 mutations (27.5%) were small deletions or duplications predicted to lead to a frameshift and premature stop codon, 14 mutations (22.5%) were nonsense mutations with a stop at this amino acid position, and 9 mutations (14.5%) were splice site mutations with a predicted effect on the splicing process (Figure 1B). In summary, 35.5% were missense mutations affecting one amino acid, whereas 64.5% were mutations that are predicted to have a larger effect on the protein, suggesting a greater influence on the severity of the disease.

In Figure 1A, all mutations detected in our cohort and their distribution across the ABCA12 protein are shown. A total of 29 of 62 mutations are located within important protein domains: the transmembrane domains 1 and 2 and the ATP-binding cassette 1 and 2. A total of 18 of the 22 missense mutations are located in one of these domains, with 14 of these occurring in the ATP-binding cassettes. This suggests that a pathogenic effect of a missense mutation on the protein function is particularly large when the missense mutation is located in a functionally important domain of the ABCA12 protein. In regions outside of these domains, only 4 missense mutations were found, along with 29 nonsense, truncating, or splice-site mutations.

The comparison with the literature showed similar results. We investigated previously published missense variants in *ABCA12*, which were classified as pathogenic for ARCI (data from HGMD^®^ Professional 2022.4). A total of 72 missense mutations have been communicated to date, including the novel missense mutations of our cohort. Although only about 35% of the amino acids in ABCA12 are located in the domains mentioned above, they contain 75% of all reported missense mutations (54 of 72).

### 3.2. Mutational Hotspots in ABCA12

To analyze possible mutational hotspots in *ABCA12*, we counted the number of alleles for each mutation in our cohort. Except for 2 mutations, the number alleles for each mutation ranged from 1 to 4. A total of 2 mutations occurred significantly more frequently: c.4541G>A, p.(Arg1514His) with 11 alleles and c.4139A>G, p.(Asn1380Ser) with 25 alleles. It must be noted that most of the patients carrying these mutations were of North African origin; therefore, a founder mutation cannot be excluded. However, the mutation c.4139A>G, p.(Asn1380Ser) was also found in patients from other origins (Table 1 and in previously published patient studies).

Furthermore, we analyzed all known amino acid changes at the positions 1514 and 1380. At position 1514, we also found the exchange from arginine to cystine in our cohort. In the literature, exchanges from arginine at position 1514 to leucine and glutamine have also been described as pathogenic. At amino acid position 1380, no additional amino acid changes have been described. All these facts are strong indications for hotspot positions in *ABCA12*.

### 3.3. Genotype–Phenotype Correlation

To examine the genotype–phenotype correlation, we analyzed the kind of mutation and the phenotype of each patient of our cohort. Detailed clinical information about the phenotype was obtained in the majority of the patients (Table 1). The phenotypes of 49 patients of our cohort could be clearly assigned to HI, CIE, or LI (Figure 1C). A total of 21 patients presented as HI, 20 patients as CIE, and 8 patients as LI (Figure 1C). The subdivision into these three main phenotypes is not always clearly feasible, since the phenotypes are sometimes overlapping. A very severe ichthyosis congenita can show similarities to HI. Some patients show characteristics of both CIE and LI. Furthermore, therapeutic treatment can alter the phenotype. This leads to the result that 15 patients of our cohort could not be classified according to the three main phenotypes. Nevertheless, clear genotype–phenotype correlations can be identified: about one third of the patients (21 of 64) in our cohort were born with HI, and most of them carry biallelic truncating mutations (Figure 1D). Some patients with HI carry one truncating mutation on one allele and a splice site or missense mutation on the other allele. In contrast, patients with CIE (20 of 64) and LI (8 of 64) did not carry truncating mutations on both alleles. Most of these patients carry missense mutations on both alleles or show the combination of a missense mutation and a truncating mutation. Taken together, a combination of a truncating mutation and a missense mutation can lead to all phenotypes, whereas two truncating mutations were only found in HI patients in our cohort. Missense mutations on both alleles mainly lead to CIE or LI. A selection of patients of our cohort with the phenotypes HI, CIE, and LI are shown in Figure 2.

## 4. Discussion

The mutational spectrum of *ABCA12* mutations in our cohort comprises 22 missense mutations (35.5%) and 40 truncating or splice-site mutations (64.5%). The percentage of missense mutations in our cohort is higher than in the cohort of Akiyama et al. [12] who found 14 missense mutations (25%) from a total of 56 homozygous or compound heterozygous mutations.

A total of 18 of 22 missense mutations of our cohort were located in one of the 2 transmembrane domains or ATP-binding cassettes, reflecting the importance of these domains for the function of the ABCA12 protein. Combining our findings with the already published mutations in the literature, about 73% of the missense mutations were located in these domains. The ATP-binding cassettes seem to be particularly important, as most missense mutations were found here. Missense variants in these conserved regions of the *ABCA12* gene limit the function of the protein and therefore lead to an ARCI phenotype. Our results support previous findings of the importance of these domains [8,12]. For classification of unknown variants, the localization inside or outside these domains should be considered, especially for missense variants. Contrarily, truncating mutations in our cohort are distributed over the entire gene. This type of mutation has a greater impact on the protein due to a truncated protein or nonsense-mediated delay and therefore results in a more severe phenotype.

In our cohort, 2 mutations occur significantly more frequently: c.4541G>A, p.(Arg1514His) in 11 alleles and c.4139A>G, p.(Asn1380Ser) in 25 alleles. In Lefévre et al. [8], the mutation p.(Asn1380Ser) was already detected in five of nine families of North African origin. Due to the Northern African origin of most of the patients carrying these mutations, a founder mutation can contribute to the higher frequency of these mutations in the cohort. However, the mutation p.(Asn1380Ser) was also found in our cohort in patients from other regions and ethnicities (Table 1). Furthermore, at amino acid position 1514, different amino acid exchanges occur. All these facts are an indication for possible hotspot positions in *ABCA12*.

Patients with HI and CIE in our cohort occur at a comparatively equivalent ratio of about one third each. LI occurs less frequently. In just under a quarter of the patients, classification as one of the main phenotypes was not possible. This can be explained by insufficient information about the phenotype or by overlapping phenotypes. Some patients show features of both CIE and LI. In severe cases, some patients cannot not be clearly assigned to HI, but they show a milder phenotype, also known as chrysalis babies [24].

Our results confirm previous findings that the complete loss of ABCA12 function due to truncating mutations on both alleles generally leads to a HI phenotype. The combination of two missense mutations can lead to all phenotypes but mainly result in CIE or LI. Three of our patients with LI carried one truncating mutation and one missense mutation. In previous publications, only missense mutations have been described in patients with LI [8,25]. Our results show that the combination of a truncating mutation and a missense mutation can also lead to an LI phenotype.

In molecular genetic diagnostics, analysis of the *ABCA12* gene may reveal only one heterozygous mutation. In cases with a CIE or LI phenotype, it should be considered that these phenotypes also occur in other genes associated with ARCI. In some of these cases, the cause of the disease may be in another gene, and the patient may additionally carry a heterozygous mutation in *ABCA12*. Therefore, if a phenotype cannot be clearly assigned to one gene, all genes associated with that phenotype should be investigated. In contrast to LI and CIE, the HI phenotype is highly specific for *ABCA12* mutations. If only one heterozygous mutation is detected, it is likely that the second mutation is present but not yet found. Larger deletions, regulatory mutations, deep intronic mutations, or complex rearrangements should be considered.

Our findings contribute to a better understanding of the pathogenesis of *ABCA12* mutations, the importance of ABCA12 domains, mutational hotspot positions, and genotype–phenotype correlations. Especially, the knowledge of the expected phenotype can improve prenatal diagnosis as well as neonatal care. However, intrafamilial phenotypic variation cannot be excluded. Furthermore, in large genes such as *ABCA12*, many variants with unclear pathogenicity are identified by molecular genetic analyses. Variants already described in patients with ARCI aid in the classification of novel variants and in establishing a correct diagnosis. The expanding of the mutational spectrum of *ABCA12* mutations in our study supports a precise molecular genetic diagnostics in ARCI. Proper genetic counseling and molecular genetic diagnostics are very important for the affected families to find a strategy for disease management. For pregnant carrier women who are at high risk of having a severely affected child, invasive procedures such as chorionic villus sampling and amniocentesis and subsequent molecular genetic diagnostics is a commonly chosen method. Prenatal testing in the first trimester using cell-free DNA (NIPT) is a non-invasive method which may also be performed more frequently in the future for monogenetic diseases. In highly consanguineous marriage populations, screening of pregnant women for autosomal recessive diseases, e.g., by NIPT, may represent a method for the early detection and prevention of recessive diseases among the population [26].To date, there is no cure for ARCI, but symptomatic treatment options are increasing. In future, gene therapy may become a promising cure for this genetic disorder.

## Figures and Tables

**Figure 1 genes-14-00717-f001:**
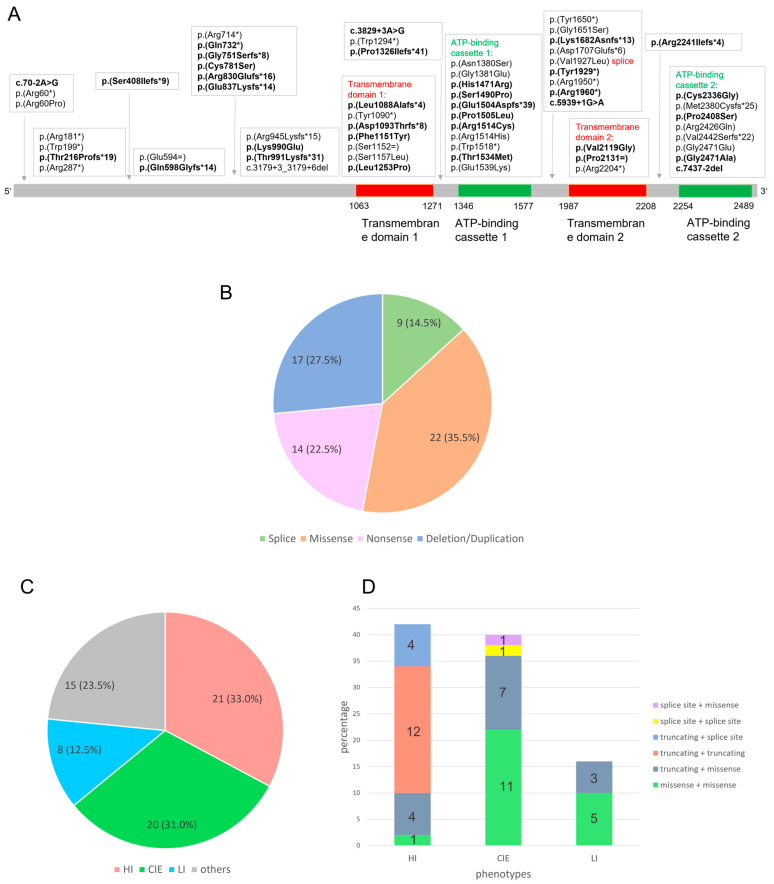
(**A**) Distribution of all mutations in the *ABCA12* gene in our cohort of 62 patients. A total of 34 novel mutations are shown in bold and 28 previously published mutations are shown in non-bold. Red: Transmembrane domain 1 and 2 in the ABCA12 protein. Green: ATP-binding cassettes 1 and 2 in the ABCA12 protein. The transmembrane domains range from amino acids 1063 to 1271 and 1987 to 2208; the ATP binding cassettes range from amino acids 1346 to 1577 and 2254 to 2489, including the AAA domains amino acids 1370 to 1554 and 2282 to 2467 (InterPro 92.0 [22], SMART 9.0 [23]). (**B**) Spectrum of mutations in *ABCA12*. In the cohort of 64 patients, 62 mutations were found: 35.5% are missense mutations, 27.5% truncating deletions/duplications, 22.5% nonsense mutations, and 14.5% splice site mutations. (**C**) Percentage distribution of phenotypes in our cohort. Patients with HI and CIE are diagnosed with greater frequency and LI is less common. The grouping of others consists of patients with unknown phenotypes or with phenotypes which cannot be clearly assigned. (**D**) Combinations of mutation classes in patients with HI, CIE, or LI in our cohort. Truncating mutations on both alleles or the combination of a truncating mutation on one allele with a splice site mutation on the other allele always led to HI in our cohort, whereas missense mutations on both alleles mainly led to CIE or LI.

**Figure 2 genes-14-00717-f002:**
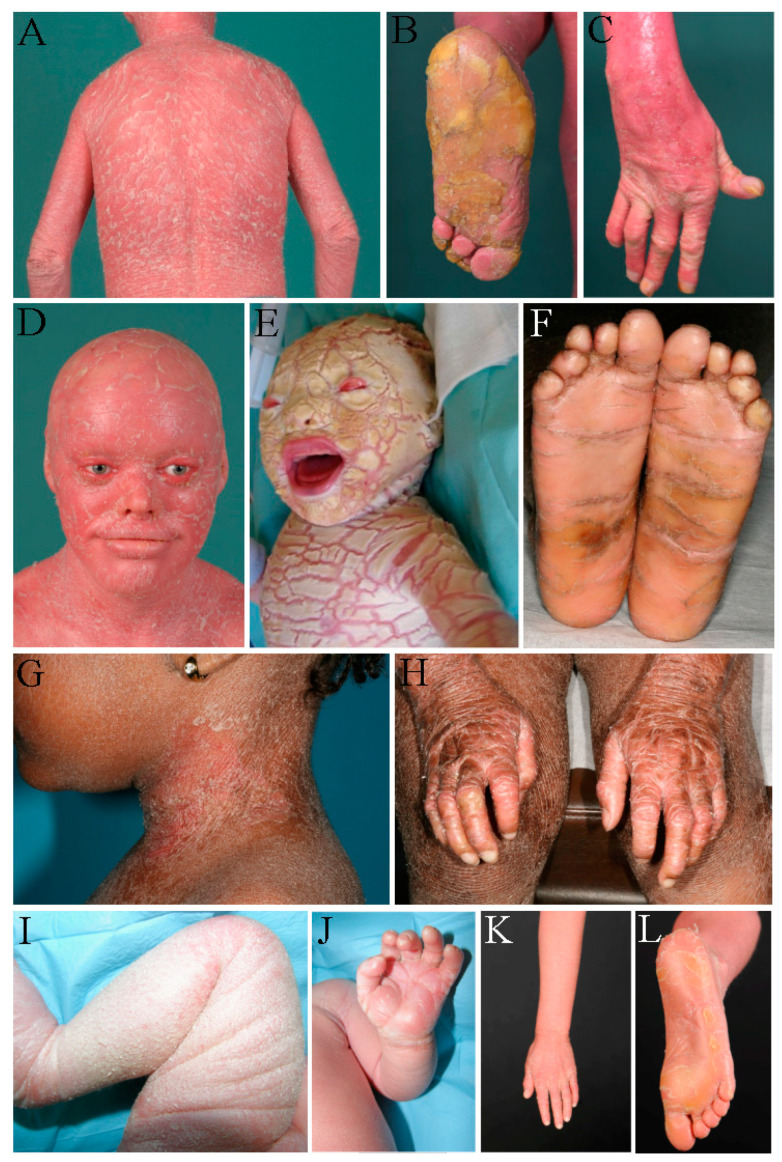
(**A**–**D**) Harlequin ichthyosis in patient 36. (**A**) Generalized dry and red scaly skin on the back, (**B**) palmoplantar hyperkeratosis, (**C**) joint contractures at fingers and thickened fingernails, and (**D**) face with red, scaly skin, ectropion, and hypoplastic ears. (**E**) Harlequin ichthyosis in patient 26 at the day of birth. Large thick plates separated by deep cracks. Ectropion, eclabion, hypoplastic nose and ears. (**F**–**H**) Lamellar ichthyosis in patient 48. (**F**) Palmoplantar hyperkeratosis, (**G**) ichthyotic skin on the neck, and (**H**) skin with large dark scales and contractures of fingers. (**I**–**L**) Congenital ichthyosiform erythroderma in patient 26. (**I**) Thickened skin with white scales in a newborn, (**J**) erythroderma of the palms, (**K**) generalized erythroderma with fine white scales, and (**L**) palmoplantar hyperkeratosis.

**Table 1 genes-14-00717-t001:** Identified mutations in *ABCA12* in patients with ARCI.

Patient	Age	Sex	Origin/Ethnicity	Consanguinity	Phenotype	Mutation 1	Mutation 2
1	n/a	m	Morocco (North African)	yes	CIE	c.4541G>A, p.(Arg1514His)	c.4541G>A, p.(Arg1514His)
2	n/a	m	Morocco (North African)	yes	CBB	c.4142G>A, p.(Gly1381Glu)	c.4142G>A, p.(Gly1381Glu)
3	33 y	m	Morocco (North African)	yes	CBB	c.4139A>G, p.(Asn1380Ser)	c.4139A>G, p.(Asn1380Ser)
4	53 y	f	France (Caucasian)	no	CBB	c.4139A>G, p.(Asn1380Ser)	**c.5878C>T, p.(Arg1960*)**
5	31 y	f	Algeria (North African)	yes	CIE	**c.4601C>T, p.(Thr1534Met)**	**c.4601C>T, p.(Thr1534Met)**
6	50 y	m	Morocco (North African)	yes	CIE	c.4615G>A, p.(Glu1539Lys)	c.4615G>A, p.(Glu1539Lys)
7	27 y	m	Algeria (North African)	yes	CBB	c.4139A>G, p.(Asn1380Ser)	c.4139A>G, p.(Asn1380Ser)
8	27 y	f	Morocco (North African)	yes	CBB	c.4139A>G, p.(Asn1380Ser)	c.4139A>G, p.(Asn1380Ser)
9	37 y	m	Algeria (North African)	no	CIE	c.4139A>G, p.(Asn1380Ser)	c.4951G>A, p.(Gly1651Ser)
10	42 y	f	Algeria (North African)	yes	CIE	c.4951G>A, p.(Gly1651Ser)	c.4951G>A, p.(Gly1651Ser)
11	45 y	f	Algeria (North African)	no	CIE	c.4139A>G, p.(Asn1380Ser)	**c.3260dup, p.(Leu1088Alafs*4)**
12	44 y	m	Algeria (North African)	yes	CIE	**c.3758T>C, p.(Leu1253Pro)**	**c.3758T>C, p.(Leu1253Pro)**
13	24 y	m	Algeria (North African)	yes	CIE	c.4139A>G, p.(Asn1380Ser)	c.4139A>G, p.(Asn1380Ser)
14	27 y	m	Tunisia (North African)	yes	CIE	c.4139A>G, p.(Asn1380Ser)	c.4139A>G, p.(Asn1380Ser)
15	20 y	f	African	yes	LI	c.4541G>A, p.(Arg1514His)	c.4541G>A, p.(Arg1514His)
16	18 y	f	African	yes	LI	c.4541G>A, p.(Arg1514His)	c.4541G>A, p.(Arg1514His)
17	17 y	m	France (Caucasian)	no	CIE	c.4139A>G, p.(Asn1380Ser)	**c.1792_1801del, p.(Gln598Glyfs*14)**
18	16 y	f	France (Caucasian)	yes	LI	c.4541G>A, p.(Arg1514His)	c.4541G>A, p.(Arg1514His)
29	n/a	n/a	Italy	yes	HI	c.859C>T, p.(Arg287*)	c.859C>T, p.(Arg287*)
20	n/a	n/a	n/a	no	HI	c.4554G>A, p.(Trp1518*)	c.4139A>G, p.(Asn1380Ser)
21	n/a	n/a	n/a	no	HI	c.7137del, p.(Met2380Cysfs*25)	c.7412G>A, p.(Gly2471Glu)
22	52 y	m	Denmark (Caucasian)	no	HI	c.596G>A, p.(Trp199*)	c.1782G>A, p.(Glu594=) splice site mutation
23	12 y	f	North African	no	HI	**c.3977del,** **p.(Ser1326Ilefs*41)**	**c.3977del,** **p.(Ser1326Ilefs*41)**
24	11 y	n/a	Germany (Caucasian)	no	HI	**c.2972_2988del, p.(Thr991Lysfs*31)**	c.5779G>T, p.(Val1927Leu) splice site
25	n/a	n/a	n/a	yes	HI	c.2486dup, p.(Arg830Glufs*16)	c.2486dup, p.(Arg830Glufs*16)
26	13 y	m	Switzerland (Caucasian)	no	CIE	**c.2968A>G, p.(Lys990Glu)**	**c.3276del, p.(Asp1093Thrfs*8)**
27	26 y	f	n/a	yes	HI	c.3270del, p.(Tyr1090*)	c.3270del, p.(Tyr1090*)
28	10 y	f	Lybia (North African)	yes	HI	**c.7222C>T, p.(Pro2408Ser)** homozygous	**c.7412G>C, p.(Gly2471Ala)** homozygous
29	9 y	m	Pakistan	yes	HI	c.7323del, p.(Val2442Serfs*22)	c.7323del, p.(Val2442Serfs*22)
30	12 y	m	Sweden (Caucasian)	no	CIE, no CBB	**c.3452T>A, p.(Phe1151Tyr)**	c.4139A>G, p.(Asn1380Ser)
31	n/a	m	Turkish	yes	HI	c.4950C>A, p.(Tyr1650*)	c.4950C>A, p.(Tyr1650*)
32	9 y	f	Turkish	no	HI	c.179G>C, p.(Arg60Pro)	c.541C>T, p.(Arg181*)
33	35 y	m	Germany (Caucasian)	no	LI with PPK	c.4541G>A, p.(Arg1514His)	c.5121_5124del, p.(Asp1707Glufs*6)
34	30 y	m	Italy	no	No CBB, erythema, translucent superficial scaling, PPK	c.179G>C, p.(Arg60Pro)	**c.4412A>G, p.(His1471Arg)**
35	9 y	f	Germany (Caucasian)	no	HI	**c.2194C>T, p.(Gln732*)**	c.3270del, p.(Tyr1090*)
36	28 y	m	Germany (Caucasian)	no	HI	**c.3829+3A>G, p.?**	**c.6722_6723del, p.(Arg2241Ilefs*4)**
37	42 y	f	Lybia (North African)	no	n/a	**c.7222C>T, p.(Pro2408Ser)**	**c.7412G>C, p.(Gly2471Ala)**
38	49 y	m	n/a	no	n/a	**c.7222C>T, p.(Pro2408Ser)**	**c.7412G>C, p.(Gly2471Ala)**
39	49 y	m	Germany (Caucasian)	no	No HI	c.2833dup, p.(Arg945Lysfs*15)	**c.4540C>T, p.(Arg1514Cys)**
40	9 y	f	Germany (Caucasian)	no	No HI	**c.646_647del, p.(Thr216Profs*19)**	c.4139A>G, p.(Asn1380Ser)
41	20 y	f	Turkish	yes	LI	c.4139A>G, p.(Asn1380Ser)	c.4139A>G, p.(Asn1380Ser)
42	8 y	f	Lebanon	yes	HI	c.3882G>A, p.(Trp1294*)	c.3882G>A, p.(Trp1294*)
43	n/a	m	Germany (Caucasian)	n/a	HI	**c.5787T>G, p.(Tyr1929*)**	**c.5787T>G, p.(Tyr1929*)**
44	29 y	m	Turkish	no	CIE	c.859C>T, p.(Arg287*)	c.4139A>G, p.(Asn1380Ser)
45	7 y	m	Germany (Caucasian)	no	CIE, mild PPK	**c.4540C>T, p.(Arg1514Cys)**	c.5848C>T, p.(Arg1950*)
46	7 y	m	Turkish	no	HI	**c.4512_4515del, p.(Glu1504Aspfs*39)**	**c.4512_4515del, p.(Glu1504Aspfs*39)**
47	17 y	m	Germany (Caucasian)	no	Mild HI	**c.2251_2252delinsT, p.(Gly751Serfs*8)**	c.3456G>A, p.(Ser1152=) splice site mutation
48	16 y	f	Senegal (African)	yes	LI	c.4541G>A, p.(Arg1514His)	c.4541G>A, p.(Arg1514His)
49	38 y	m	France (Caucasian)	no	CIE	c.178C>T, p.(Arg60*)	c.4139A>G, p.(Asn1380Ser)
50	40 y	f	France (Caucasian)	no	No HI, arthrogryposis	**c.2509del, p.(Glu837Lysfs*14)**	c.4139A>G, p.(Asn1380Ser)
51	7 y	m	Germany (Caucasian)	no	CIE	c.3179+3_3179+6del, p.?	**c.7437-2del, p.?**
52	7 y	m	n/a	yes	HI	c.6610C>T, p.(Arg2204*)	c.6610C>T, p.(Arg2204*)
53	14 y	m	Germany (Caucasian)	no	CIE with PPK and LI. CBB at birth	**c.70-2A>G, p.?**	c.7277G>A, p.(Arg2426Gln)
54	16 y	f	Germany (Caucasian)	no	LI	c.2140C>T, p.(Arg714*)	**c.4514C>T, p.(Pro1505Leu)**
55	5 y	m	Germany (Caucasian)	no	CIE	c.4139A>G, p.(Asn1380Ser)	**c.4468T>C, p.(Ser1490Pro)**
56	47 y	m	Germany (Caucasian)	no	LI	**c.1221dup, p.(Ser408Ilefs*9)**	c.3470C>T, p.(Ser1157Leu)
57	30 y	m	Germany (Caucasian)	no	HI	c.2140C>T, p.(Arg714*)	**c.2341T>A, p.(Cys781Ser)**
58	31 y	f	Germany (Caucasian)	no	Mild ichthyosis	c.4139A>G, p.(Asn1380Ser)	**c.4540C>T, p.(Arg1514Cys)**
59	4 y	f	Germany (Caucasian)	no	CBB	**c.5939+1G>A, p.?**	**c.7006T>G, p.(Cys2336Gly)**
60	22 y	m	Germany (Caucasian)	no	CIE, no CBB	c.4139A>G, p.(Asn1380Ser)	**c.6393G>T, p.(Pro2131=)** splice site mutation
61	3 y	m	Serbia/Hungary	no	Ichthyosis congenita gravis	c.179G>C, p.(Arg60Pro)	**c.6356T>G, p.(Val2119Gly)**
62	45 y	f	Germany (Caucasian)	no	Mild CIE, without therapy lamellar desquamation, PPK	c.2833dup, p.(Arg945Lysfs*15)	**c.4540C>T, p.(Arg1514Cys)**
63	1 y	f	Germany (Caucasian)	n/a	Severe CBB	c.3470C>T, p.(Ser1157Leu)	c.3470C>T, p.(Ser1157Leu)
64	1 y	m	Syria (North African)	yes	HI	**c.5046_5050del, p.(Lys1682Asnfs*13)**	**c.5046_5050del, p.(Lys1682Asnfs*13)**

In bold: novel mutations. CBB: collodion baby, PPK: palmoplanar keratosis, HI: harlequin ichthyosis, CIE: congenital ichthyosiform erythroderma, LI: lamellar ichthyosis, y: years.

## Data Availability

Data is contained within the article or Appendix A.

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
