# Peer review of "Mutational Spectrum of the ABCA12 Gene and Genotype–Phenotype Correlation in a Cohort of 64 Patients with Autosomal Recessive Congenital Ichthyosis"

_genes, 2023, doi:10.3390/genes14030717_

Round 1
Reviewer 1 Report
Hotz et al. have presented a study identifying 34 novel mutations in ABCA12 that are associated with autosomal recessive congenital ichthyosis, which builds on pre-existing literature linking biallelic mutations in this gene to this condition. In doing so, they characterized mutational hotspots in ABCA12 that are most associated with ARCI, and described variant classes that lead to more severe disease phenotypes, particularly when the mutation disrupts ABCA12's functional domains. The study provides interest, especially to the ARCI community.
With respect to the methods, it makes sense why different sequencing methods were used for different patients due to the timespan in which blood samples were collected. However, I would like to see some discussion of how different sequencing methods may impact the ability to detect different classes of mutations. For example, is there a potential bias for splice site mutations being missed in older samples which were done with Sanger sequencing? In future studies of ARCI, would there be merit in resequencing patient samples using more contemporary methods, like long read technologies?
Some specific comments:
1) For ease of comparison, pie charts in Figure 1B and 1C can be replaced with bar plots.
2) Figure 1D would be better represented as relative abundances of mutation types, with total number of patients added as bar labels.
3) Figure 2J, K, and L appear to be cut off. (This may just be a formatting issue)
I noticed a few typos.
Line 45, "We reveal the possible"
Line 113, "to date" is used twice in the same sentence.
Line 282, should read "the patient is additionally a carrier"
Author Response
Referee 1:
- With respect to the methods, it makes sense why different sequencing methods were used for different patients due to the timespan in which blood samples were collected. However, I would like to see some discussion of how different sequencing methods may impact the ability to detect different classes of mutations. For example, is there a potential bias for splice site mutations being missed in older samples which were done with Sanger sequencing? In future studies of ARCI, would there be merit in resequencing patient samples using more contemporary methods, like long read technologies?
Reply: Thank you for your remark. In all patients of our cohort, all coding exons and flanking sequences (+/- 20 base pairs) of the gene was analyzed. We have added this information in the methods. Deep intronic variants cannot be detected either with Sanger sequencing or with the NGS methods used. In patients with only one mutation, the second may not be detected due to technical limitations. In these patients, however, it cannot be ruled out that the patients are only carriers and that the cause of the ichthyosis lies in another gene. In our cohort, we present only patients with biallelic mutations in ABCA12. Therefore, the discussion about limitations of our Sequencing methods seem not so appropriate to us.
- For ease of comparison, pie charts in Figure 1B and 1C can be replaced with bar plots.
Reply: Thank you for your remark. We have tested the proposed bar plots in Fig. 1B and 1C but we think the bar plots don’t look so good. We have discussed different illustrations of Fig. 1B and 1C, but at last we prefer pie charts in these figures, because we think they show the proportions more clearly. We hope this representation is sufficient for you.
- Figure 1D would be better represented as relative abundances of mutation types, with total number of patients added as bar labels.
Reply: Thank you for your remark, we have changed Fig. 1D. The Y axis is now showing relative abundances.
- Figure 2J, K, and L appear to be cut off. (This may just be a formatting issue)
Reply: Thank you for your remark. Indeed, the Fig. 1D was cut off due to formatting by the editorial office. We will forward the information to the editorial office.
- Line 45, "We reveal the possible"
Line 113, "to date" is used twice in the same sentence.
Line 282, should read "the patient is additionally a carrier"
Reply: Thank you for this information. We have changed these mistakes.
Reviewer 2 Report
Alrun Hotz et al. reported Mutational Spectrum of the ABCA12 Gene and Genotype-Phenotype-Correlation in a Cohort of 64 Patients with Autosomal Recessive Congenital Ichthyosis.
The clinical and molecular data very are very interesting and could be of interest to the readers. However, there are some comments that need to be addressed.
General comments:
+ Follow nomenclature: https://varnomen.hgvs.org/
*- The reference sequence (NP_) and NM_ should also be added.
**Gene name should be in italics. Kindly check.
**USE (OMIM #), and use the proper name of the disorder.
--Spacing issues need attention.
++English grammar should be improved.
Appropriate
Methods/Results
**IRB approval number should be mentioned.
Consanguineous parents?? Or Not?
Add subheadings…
Results
How the variants were classified as pathogenic ACMG criteria fulfilled?
**Was the patient treated with any medication? Condition of patients before and after treatment?
Make a table showing pathogenicity of all the variants identified and ACMG classification.
Discussion
Any genotype-phenotype correlation? Location of variants associated with variable phenotypes?
In the last discussion, add lines for future perspectives; discuss newborn screening, NIPT, PGT-A, and PGT-M, for example.:
Rare genetic disease prevention strategies and the future of gene therapy should be discussed.
Proper genetic counseling for the affected family is essential in the case of rare genetic diseases. Furthermore, parenteral genetic screening/diagnosis is the best strategy for managing this disease, which currently has no therapy (Alfadhel et al., 2019; Alyafee, Al Tuwaijri, et al., 2021; Alyafee, Alam, et al., 2021). Reporting additional cases associated with this gene would help identify genotype–phenotype correlations and lead to clinical trials in the future (Alfadhel et al., 2021).
https://doi.org/10.1002/acn3.50898
https://doi.org/10.3389/fgene.2021.630787
https://doi.org/10.3390/genes12040461.
doi: 10.3389/fgene.2022.1047474.
Author Response
Referee 2:
- Follow nomenclature: https://varnomen.hgvs.org/
Reply: Our nomenclature corresponds to the current nomenclature in varnomen and was applied consistently.
- The reference sequence (NP_) and NM_ should also be added.
Reply: We have added this information in the methods.
- Gene name should be in italics. Kindly check.
Reply: Thank you for your remark. We have checked the text, we made corrections in line 44, line 96 and line 192. The gene names are in italic, the protein names in non-italic.
- USE (OMIM #), and use the proper name of the disorder.
Reply: We have added this information in the introduction.
- Spacing issues need attention.
Reply: We have checked the spaces in the manuscript. We made corrections in line 64.
- English grammar should be improved.
Reply: Thank you for your remark. A native speaker have checked our manuscript and we made several improvements.
- IRB approval number should be mentioned.
Reply: We have added the IRB number at the end of the manuscript.
- Consanguineous parents?? Or Not?
Reply: Thank you for the remark. We have added this information in Table 1 in an additional text column.
- Add Subheadings
Reply: Thank you for this remark. Since the method part is short, we don't think subheadings don’t make much sense.
- How the variants were classified as pathogenic ACMG criteria fulfilled? Make a table showing pathogenicity of all the variants identified and ACMG classification.
Reply: Information about classification of the novel variants can be found in Table S1 (Supplementary material). They were already listed there when we submitted the manuscript. Maybe Table S1 not have been sent to you.
- Was the patient treated with any medication? Condition of patients before and after treatment?
Reply: We did not study medical treatments of our patients, since the focus of our study is the mutational spectrum and genotype-phenotype-correlation. Furthermore, we did not collect detailed information about the medical treatments of our patients.
- Any genotype-phenotype correlation? Location of variants associated with variable phenotypes?
Reply: We have studied the main phenotypes associated with ARCI due to different mutation classes and locations in the gene ABCA12. We did not investigate single variants, because interpretation of phenotypes is difficult due to the combination of these variants with other variants (except homozygous variants). Therefore we focussed on combination of mutation classes (truncating, missense…).
- Discussion
Reply: We have expanded the discussion with the suggested themes.